# Fault Diagnosis of Permanent Magnet DC Motors Based on Multi-Segment Feature Extraction

**DOI:** 10.3390/s21227505

**Published:** 2021-11-11

**Authors:** Lixin Lu, Weihao Wang

**Affiliations:** School of Mechatronic Engineering and Automation, Shanghai University, 99 Shangda Road, BaoShan District, Shanghai 200444, China; lulixin@shu.edu.cn

**Keywords:** fault diagnosis, permanent magnet DC motor, support vector machine, classification and regression tree, *k*-nearest neighbor, feature extraction

## Abstract

For permanent magnet DC motors (PMDCMs), the amplitude of the current signals gradually decreases after the motor starts. Only using the signal features of current in a single segment is not conducive to fault diagnosis for PMDCMs. In this work, multi-segment feature extraction is presented for improving the effect of fault diagnosis of PMDCMs. Additionally, a support vector machine (SVM), a classification and regression tree (CART), and the *k*-nearest neighbor algorithm (*k*-NN) are utilized for the construction of fault diagnosis models. The time domain features extracted from several successive segments of current signals make up a feature vector, which is adopted for fault diagnosis of PMDCMs. Experimental results show that multi-segment features have a better diagnostic effect than single-segment features; the average accuracy of fault diagnosis improves by 19.88%. This paper lays the foundation of fault diagnosis for PMDCMs through multi-segment feature extraction and provides a novel method for feature extraction.

## 1. Introduction

Permanent magnet DC motors (PMDCMs) are widely used in automobiles, household electrical appliances, industrial production, and so on. PMDCM faults can bring in additional noise, harmful vibration and equipment shutdown, which will result in lots of economic losses and poor user experience. Thus, it is necessary to carry out comprehensive fault diagnosis for PMDCMs before they leave the factory. However, artificial diagnosis easily leads to misjudgment since the current amplitude of PMDCMs gradually decreases after the motor starts. Note that artificial intelligence (AI) is an effective way to solve this problem.

Machine learning (ML) has been widely utilized for fault diagnosis, which involves industry, medical, and many other fields [1,2,3]. Previous studies show that feature engineering is of great importance in ML application. Thus, feature extraction plays an important role in motor fault diagnosis. Most studies for motor fault diagnosis focus on time domain, frequency domain, and time-frequency domain features of vibration or current signals [1,3]. The difficulty of feature extraction lies in finding the features that can represent the original signal attribute [3]. Li et al. [4] presented a composite fault diagnosis method for rolling bearing by means of compressed sensing framework, which can effectively extract fault characteristics. Plaza et al. [5] applied four feature extraction methods to surface roughness prediction. The results show that singular spectrum analysis and the wavelet packet transform perform better than time-direct analysis and power spectral density. Ding et al. [6] presented a transient feature extraction method of encoder signal for condition assessment of planetary gearboxes under variable rotational speed, which was confirmed by simulation and case study. Shi et al. [7] presented a feature extraction method based on the fractional Fourier transform, which was utilized to estimate the load of a tubular ball mill. Cai et al. [8] applied sparsity-enabled signal decomposition for fault feature extraction of the gearbox, which outperformed empirical mode decomposition and spectral kurtosis. Li et al. [9] proposed a novel feature extraction method by combining the advantages of the empirical wavelet transform and reverse dispersion entropy, which outperformed the other five latest feature extraction technologies. Lin et al. [10] presented a novel feature extraction model based on the concept of the discriminative graph signal, which outperformed the existing methods. Yang et al. [11] presented a Bag-of-Words-based feature extraction method for change detection in the rotational speed of industrial machinery, which had better robustness than the state-of-the-art competitors. Chen et al. [12] reviewed and categorized most based on the existing data-driven detection and diagnosis of faults (FDD) mainstream methods for traction systems in high-speed trains and summarized the challenges of successful implementations of FDD on practical high-speed trains. Benefiting from theoretical developments of data-driven FDD strategies, an explosive growth of data-driven FDD methods would be witnessed in the coming years. Azamfar et al. [13] utilized the motor current signature by multi-sensor data fusion for gearbox fault detection and diagnosis based on a 2-D convolutional neural network without any need for manual feature extraction. The performance of the proposed method was evaluated in two different experimental studies. Compared with some of the well-known ML methods, the proposed method exhibits the best performance in terms of classification accuracy. Juez-Gil et al. [14] proposed a new technique based on multiple sensor information for early diagnosis of faulty conditions in induction motors. The principal component analysis (PCA) utilized to reduce the number of multiple sensor information features, and the effectiveness of the proposed method is verified by using a complete set of experimental data. Glowacz [15] proposed a fault diagnosis technique of three-phase induction motors by using acoustic signals and proposed two feature extraction methods: SMOFS-32-MULTIEXPANDED-2-GROUPS and SMOFS32-MULTIEXPANDED-1-GROUP. The *k*-NN, back propagation neural network (BPNN), and modified classifier based on words coding were used for the recognition of acoustic signals. Widodo et al. [16], based on transient signal, presented a method for induction motor fault diagnosis by using component analysis and an SVM. The start-up transient current signal was used for fault diagnosis by extractive features. Moreover, PCA is also used to study the influence of the number of features towards the classification accuracies (training and testing). It should be noted that application of these feature extraction methods is mainly focused on the case that the original signal is relatively stable. However, these methods may not be suitable for the non-stationary current signals of PMDCMs.

Previous studies have focused on feature transform and feature compression, such as PCA, ICA, and so on. Some scholars have focused on the innovation of the algorithm. Others have used multi-sensors, which can obtain more signals for the same object. Having the best performance of fault diagnosis, the classifier can approach the best performance infinitely; thus, finding suitable features that can reflect the attributes of the signal seems so important. Time domain, frequency domain, and time-frequency domain can only represent the local features of signals. To overcome the shortcoming of feature extraction and improve the fault diagnosis performance of PMDCMs, this paper presents a new multi-segment feature extraction method based on the time domain features. The time domain features of current signals in several successive segments are put together to make up a feature vector, which is adopted as the input of fault diagnosis. A support vector machine (SVM) [17,18], classification and regression tree (CART) [18], and *k*-nearest neighbor (*k*-NN) [19] are utilized for the construction of the fault diagnosis model. In this work, an experimental platform is set up for proving the effectiveness of multi-segment feature extraction.

This paper is organized as follows: Section 2 introduces the backgrounds of the SVM classifier, CART classifier, *k*-NN classifier, $k^{\prime }$-fold cross-validation [20,21] theory, and feature extraction; Section 3 introduces the proposed multi-segment feature extraction method; the experimental platform, data collection, and experimental results are given in Section 4; performance discussion and future research focuses are provided in Section 5; and, finally, Section 6 concludes this paper.

## 2. Backgrounds

### 2.1. Classifier

A support vector machine (SVM) [17,18], classification and regression tree (CART) [18], *k*-nearest neighbor algorithm (*k*-NN) [19] are utilized to verify the effectiveness of the proposed multi-segment feature extraction method.

#### 2.1.1. SVM

An SVM realizes binary classification through data interval maximization in input space or feature space. SVMs can be divided into linear-separable SVM, linear SVM, and nonlinear SVM. When the training samples are linearly separable in input space, the linear-separable SVM can find a hyper-plane with maximum interval to classify these data. In most cases, training samples are linearly inseparable in input space, and the nonlinear SVM can find a hyper-plane with maximum interval to classify these data by mapping them into a feature space. This process is called kernel transformation. The most widely-used kernel function in SVMs contains a polynomial function, a Gaussian function, and a sigmoid function. In this work, the Gaussian function is adopted as the kernel function. The penalty parameter C and the kernel parameter $\sigma^{2}$ need to be confirmed by the SVM with the Gaussian kernel function.

In this paper, the grid-search method is utilized to determine the two model parameters (C and $\sigma^{2}$) of the SVM. Optimization of the model parameters in the SVM is carried out in a two-dimensional space: $\left(C_{i},\;\sigma_{i}^{2}\right)\in \left[2^{-8},\;2^{8}\right]\times \left[2^{-8},\;2^{8}\right]$. The grid size $∆C_{i}$ and $∆\sigma_{i}^{2}$ is set to $2^{1}$ and $2^{1}$, respectively. In the grid-search method, intersections in the grid space are formed by the combination of $C_{i}$ and $\sigma_{i}^{2}$. Each intersection corresponds to a fitness.

The SVM belongs to binary classifier. The One-versus-one (OVO) or one-versus-rest (OVR) method can be utilized to realize the multi-category classification. In this work, the OVO method is utilized to realize multi-category classification, in consideration of the sample size and limited categories.

#### 2.1.2. CART

Classification and regression have the same essence for CART. The feature space is gradually divided from the root node until the leaf node gives the prediction result. The classification tree gives discrete values, while the regression tree gives continuous values. The classification tree selects the characteristics of the decision tree based on the Gini-index, and the regression tree selects the segmentation points based on square error.

The CART should discretize the continuous-value attributes, divide each possible value a of the existing feature A into positive and negative, and at the same time calculate the Gini-index. The feature with the smallest Gini-index was selected as an optimal divide feature, which divided the training set into two training subsets. The Gini-index should be circularly calculated, and the dataset should be circularly segmented until data samples are below than the data samples’ threshold value, or Gini-index is less than the Gini-index threshold value, or there are no more features.

#### 2.1.3. *k*-NN

The *k*-NN algorithm is a clustering algorithm; the main idea of the *k*-NN algorithm is that each sample can be represented by its nearest k neighbors. Most of the nearest k samples belong to a certain class, so the sample belongs to the class. The *k*-NN algorithm is a lazy algorithm, which can predict classes without model training. The main steps of the *k*-NN algorithm are exhibited as follows:
Step 1:Normalize the feature to interval [0, 1], according to Equation (1) [19];
(1)$$x^{\prime }=\frac{x-x_{min}}{x_{max}-x_{min}}$$

Step 2:Calculate the distance between the sample to be tested and all training samples. In this paper, Euclidean distance is used, which is given by [19]:


(2)
$$d\left(x,y\right)=\sqrt{\overset{n}{\underset{j=1}{\sum }}{\left(x_{j}-y_{j}\right)}^{2}}$$


Step 3:Sort the Euclidean distance in descending order that is calculated in step 2;Step 4:Utilize the majority vote for the first k-sorted training samples according to the Euclidean distance, and the class of the sample to be tested is the one with the largest number of votes.

### 2.2. $k^{\prime }$-Fold Cross-Validation

The procedure of $k^{\prime }$-fold cross-validation is utilized to ensure the stability of the model results. Firstly, the training dataset is averagely divided into $k^{\prime }$ subsets. Secondly, the fault diagnosis model is trained and verified (i.e., model construction and evaluation) for $k^{\prime }$ iterations. In the i th iteration, the i th subset is adopted as the test set, while the remaining $k^{\prime }-1$ subsets are adopted as the training set. The accuracy rate of the model is adopted as the evaluation indicator. Thirdly, the average of the $k^{\prime }$ indicators in the iterations is considered the fitness. The $k^{\prime }$-fold cross-validation procedure is shown in Figure 1.

### 2.3. Feature Extraction

Most of the time, the features of raw signals are adopted as the input of machine learning (ML) since the raw signals contain lots of redundant information. However, with the increase in storage space and the enhancement of data processing ability, as a branch of ML in deep learning (DL), models’ raw time series signals are indeed used as inputs. Consider the efficiency of the training model; when the training samples are not enough to train DL model, the other ML methods play a great role. The input of the other ML methods is the features extracted from the raw signal. Objectively speaking, feature extraction plays an important role in ML application. The extracted features should retain the characteristics of the raw signals as much as possible. Time domain features, frequency domain features, and time-frequency domain features can reflect the features of signals. This paper proposes a novel feature extraction method based on a comparison of the multi-segment feature extraction method with the single-segment feature extraction method. Therefore, in this work, nine time domain features are extracted for motor fault diagnosis, as shown in Table 1.

## 3. Proposed Multi-Segment Feature Extraction Method

Anmpulse current will be generated as soon as the PMDCMs are powered on, and then it will gradually reach a stable level. As shown in Figure 2, the current signals of each type of PMDCM are different in both starting and stable stages. To distinguish the fault type of PMDCMs effectively, the time domain features as listed in Table 1 are extracted from several successive segments of current signals, respectively, and the extracted signal features make up a feature vector, which is adopted for fault diagnosis of PMDCMs.

The impulse current signal is divided into multiple segments for feature extraction. After being powered on for 1 s, the current of PMDCMs tends to be stable and have a steady downward trend; therefore, a 2 s length of current signal can reflect the fault of PMDCMs. In order to collect the complete start current signal of PMDCMs, the start time of the signal acquisition card is 0.5 s earlier than the start time of PMDCMs. In the time from 0.5 s to 1 s, the current signal of PMDCMs changes obviously, and between different faults of PMDCMs, the current signal is distinguished obviously. Therefore, the current signal from 0.5 s to 1 s was divided into multi-segment and extract features.

As shown in Figure 2, the current signals (0.5~2 s) of PMDCMs are divided into eight segments, i.e., A, B, C, D, E, F, G, and H. Among them, A, B, C, D, E, and F are the starting stages of the current signal, with a duration of 0.05 s. Additionally, G and H are the stable stages of the current signal, with a duration of 0.5 s. The nine time domain features (see Table 1) are extracted from the eight segments of current signals, respectively. A total of 8×9=72 signal features can be extracted from each data file (i.e., data sample) and make up a feature vector. This process is referred to as multi-segment features extraction.

To optimize the size and number of the signal segments, seven kinds of multi-segment feature extraction strategies are utilized in this paper, as shown in Table 2.

To verify the validity of the proposed multi-segment feature extraction, which is used to fault diagnosis of PMDCMs, the SVM, CART, and *k*-NN were utilized to construct the fault diagnosis model separately. The procedure of fault diagnosis of PMDCMs based on multi-segment feature extraction by the SVM, CART, and *k*-NN can be seen in Figure 3. The main steps of fault diagnosis of PMDCMs based on the SVM, CART, and *k*-NN by multi-segment features was as follows:Step 1:The PMDCMs’ current signal acquisition platform was established; for more details, see Figure 4 and Section 4.1;Step 2:A total of 1200 sets of PMDCMs current signals were collected; more details are given in Section 4.2;Step 3:The current signals of PMDCMs were divided into eight segments; details of the segment method are given in Section 3;Step 4:Extract the time domain features (see Table 1) of multi-segment signals, which are made up of feature vectors;Step 5:The SVM-based fault diagnosis models of PMDCMs were trained with the training set. Two parameters (the penalty parameter C
and the kernel parameter $\sigma^{2}$) affected the performance of the Gaussian kernel-based SVM. Thus, the $k^{\prime }$-fold cross-validation and grid-search method were utilized for optimizing the penalty parameter C and the kernel parameter $\sigma^{2}$. After the penalty parameter C and the kernel parameter
$\;\sigma^{2}$ optimization, the model’s performance was assessed independently in the corresponding test set. For more details, please see Section 4.3 and Figure 5;Step 6:A total of 1200 samples were divided into $k^{\prime }$ equal size partitions randomly. The 1200 samples were randomly sorted and then numbered from 1 to $k^{\prime }$. One group of samples with the same number (i.e., subset) was used as the test set, and the other samples were used as the training set. This procedure repeats $k^{\prime }$ times by taking each subset as the test set in turn. This constructs the CART and *k*-NN-based fault diagnosis models of PMDCMs, respectively. For more details, please see Section 4.3.

## 4. Experimental Results and Analysis

This study aims at constructing an effective and feasible fault diagnosis model for permanent magnet DC motors (PMDCMs) by means of multi-segment feature extraction.

### 4.1. Experimental Introduction

The left half of the Figure 4 shows the data acquisition platform, which consists of a personal computer (PC), a control cabinet, and PMDCMs. The PC is mainly used for data storage, data processing, results display, and so on. The detail structure of the control cabinet is shown in the right half of Figure 4. It consists of a 24 V power supply, 12 V power supply, front-end relay, timer relay, 4-channel relay, sampling resistance with high frequency, and a data acquisition card. The type and function of these apparatuses are listed in Table 3. The 24 V power supply provides the power source for the relays and data acquisition card. To eliminate the cross effect of current signals, one 12 V power supply provides the power source for only one PMDCM. The 4-channel relay serves to control the power on and off for the PMDCMs. The timer relay serves to delay the power on of the circuit, which aims at collecting the current signals of the PMDCMs when powered on. The intermediate relay can be triggered by 5 V input voltage and controls the power on and off for the timer relay. The data acquisition card serves to monitor the starting signal and collect the current signals of PMDCMs.

The studied PMDCMs fall under six categories: healthy, slight noise, loud noise, harsh noise, shaft unbalance, and bearing slipping, as shown in Table 4.

### 4.2. Data Collection

In this work, the collected current signals of PMDCMs are graphically shown in the PC terminal (see Figure 2). Sampling frequency and sampling duration of the current signals of PMDCMs are set to 10 kHz and 5 s, respectively. The current signals are collected by data acquisition card and saved in LVM format through the software LabVIEW. A collection of 200 data files (i.e., data samples) is obtained from each type of PMDCM. Thus, a total of 200×6=1200 data files are obtained in the stage of data collection.

### 4.3. Performance of Multi-Segment Features Model

A total of 1200 data files are obtained in the stage of data collection (see Section 4.2). Additionally, the feature vector is extracted from each data file in the stage of feature extraction (see Section 3). Thus, a total of 1200 feature vectors can be obtained, which are randomly divided into two subsets of equal size: the training dataset and test dataset.

In this work, in order to prove the effectiveness of the proposed multi-segment feature extraction method for fault diagnosis of PMDCMs, the SVM, CART, and *k*-NN are utilized to construct the fault diagnosis model of PMDCMs based on the training dataset. The corresponding test dataset is adopted to verify the effectiveness of the fault diagnosis model.

As for the SVM-based fault diagnosis model of PMDCMs, there are two model parameters that directly affect the performance, i.e., the penalty parameter C and the kernel parameter $\sigma^{2}$. The procedure of the parameter optimization in SVM for fault diagnosis of PMDCMs is illustrated in Figure 5.

In this work, sevenfold cross-validation is selected. The intersection that corresponds to the minimum fitness contains the two optimized model parameters ($C_{*}=8$ and $\sigma_{*}^{2}=0.0156$). Then, the SVM-based fault diagnosis model can be constructed by using the training dataset and the determined model parameters ($C_{*}=8$ and $\sigma_{*}^{2}=0.0156$). The test dataset is adopted to evaluate the performance of the constructed SVM-based fault diagnosis model, as shown in Table 5. The accuracy of the constructed SVM-based fault diagnosis model using the multi-segment features (the S005 feature set) reaches up to 93.83%.

As for the CART-based fault diagnosis model of PMDCMs, sevenfold cross-validation is selected. (Typically, sevenfold cross-validation can produce better and more stable results, as we can see in Figure 6. A total of 1200 feature vectors are randomly divided into seven subsets on average. Six subsets are adopted as the training set, and the remaining one subset is taken as the test set. The accuracy rate of the CART-based fault diagnosis model is adopted as the evaluation indicator, as shown in Table 5. The accuracy of the constructed sevenfold cross-validation CART-based fault diagnosis model using the multi-segment features (the S0025 feature set) reaches up to 89.41%.

As for the *k*-NN-based fault diagnosis model of PMDCMs, the grid-search method is utilized to determine the two model parameters. The k of the *k*-NN ranges from 3 to 11 with step size 1. The $k^{\prime }$ of $k^{\prime }$-fold cross-validation ranges from 3 to 11 with step size 1, respectively. The accuracy of the constructed *k*-NN-based fault diagnosis model with cross-validation using the multi-segment features (the S005 feature set) is shown in Table 6. It is apparent that when the k of *k*-NN is set to 5 always can perform best, and also when $k^{\prime }$ of $k^{\prime }$-fold cross-validation is set to 7, most times (5/9), it can perform better. See Figure 6 for visual illustration of the selection process regarding k and $k^{\prime }$. The accuracy of the constructed sevenfold cross-validation *k*-NN-based (k=5) fault diagnosis model using the multi-segment features (the S005 feature set) reaches up to 92.5%. Naturally, in next section (Section 4.4), the *k*-NN based fault diagnosis model of PMDCMs utilizing single-segment features (nine-dimension) is constructed with the k of *k*-NN being set to 5 and the $k^{\prime }$ of $k^{\prime }$-fold cross-validation being set to 7.

From F1 to F9, some of the features are correlated, so each feature is extracted separately in each single segment, and the SVM-, CART-, k-NN-based models are established by utilizing the eight-dimension features. The performances of the SVM-, CART-, k-NN-based models utilizing the F1 to F9 feature sets separately are shown in Table 7, the models utilizing the F1 feature set achieve the best performance one time, and the models utilizing the F2 feature set achieve the best performance time times. Therefore, F2 was used as the benchmark feature, and the other eight features and calculated Pearson correlation coefficients [22] with F2, respectively, are shown in Table 8. The Pearson correlation coefficients of F1 and F3 are more than 0.5 with F2; thus, the No_F1 and No_F3 feature sets are formed by eliminating the F1 and F3 from the S005 feature set, respectively; the No_F1_F3 feature set is formed by eliminating the F1 and F3 from the S005 feature set at the same time. The performances of the SVM-, CART k-NN-based models utilizing the No_F1, No_F3, and No_F1_F3 are shown in Table 9. Whether F1 or F3 are eliminated or F1 and F3 are eliminated at the same time, the performances of the model are not as strong as the performances of the model using the S005 feature set.

The three fault diagnosis models of PMDCMs is aimed at proving the effectiveness of the multi-segment features extraction method. As we can see in Table 10, the accuracy of the fault diagnosis model using the multi-segment (S005 feature set) features reach up to 93.83%, 89.16%, 92.5%, respectively, for more a intuitive explanation and deeper understanding of the effect of the proposed multi-segment feature extraction method. In this paper, 1200 samples are divided into two groups. One is used to train and the other is used to test. Figure 7 shows the confusion matrices of three fault diagnosis models, and the *x*-axis corresponds to the predicted status, while the *y*- represents the actual status. In Figure 7a, the confusion matrix utilizing the SVM-based ($C_{*}=8$ and $\sigma_{*}^{2}=0.0156$) fault diagnosis model of PMDCMs is shown. In Figure 7b, the confusion matrix utilizing the CART-based fault diagnosis model of PMDCMs is shown. In Figure 7c, the confusion matrix utilizing the k-NN-based fault diagnosis model of PMDCMs is shown. It can be seen from the three confusion matrices that most samples were correctly classified; however, they still have a small mounts class 5 predicted as class 1 or class 2, class 1 predicted as class 1, and class 2 predicted as class 1. As shown in Table 4, class 1 represents healthy, class 2 represents slight noise, and class 5 represents shaft unbalance. There is no obvious difference between class 1 and class 2; shaft unbalance only produces slight vibration and no noise, so class 5 is close to class 1.

### 4.4. Performance of Single-Segment Features Model

To show the effectiveness of the multi-segment features (seventy-two-dimension), the single segment (i.e., A, B, C, D, E, F, G, and H) features (nine-dimension), as shown in Figure 2, are used for model construction and evaluation, respectively. The detailed process is the same as that in Section 4.3. The testing results for each segment are listed in Table 10.

The best prediction performance of the SVM-based fault diagnosis model reaches up to 81.76% based on H-segment features, followed by 79.33% based on G-segment features. The best prediction performance of the CART-based fault diagnosis model reaches up to 73.67% based on G-segment features, followed by 72.92% based on H-segment features. The best prediction performance of the *k*-NN-based fault diagnosis model reaches up to 76.25% based on H-segment features, followed by 72.42% based on G-segment features. The SVM-, CART-, and *k*-NN-based models all achieved better prediction results on G or H single-segment features than A, B, C, D, E, and F single-segment features. The reason for this might be not only that G and H segments have longer durations than A, B, C, D, E, and F segments but also that in the stable stage, they can have more stable features.

It can be seen from the data in Table 10 that the accuracy of the constructed SVM-based fault diagnosis model using the single-segment features (nine-dimension) varies from 68.67% to 81.67%, and the average accuracy is 75.25%. It can be found that the SVM-based fault diagnosis model using multi-segment features (93.83%) has a better diagnostic effect than using single-segment features (68.67~81.67%), and it increased by 18.58% on average. The accuracy of the constructed CART-based fault diagnosis model using the single-segment features (nine-dimension) varies from 68.08% to 73.67%, and the average accuracy is 70.57%. It can be found that the CART-based fault diagnosis model using multi-segment features (89.16%) has a better diagnostic effect than using single-segment features (68.08~73.67%), and it increased by 18.59% on average. The accuracy of the constructed *k*-NN-based (k=5) fault diagnosis model using the single-segment features (nine-dimension) varies from 64.67% to 76.25%, and the average accuracy is 70.04%. It can be found that the *k*-NN-based (k=5) fault diagnosis model using multi-segment features (92.5%) has a better diagnostic effect than using single-segment features (64.67~76.25%), and it increased by 22.46% on average. Fault diagnosis of PMDCMs based on multi-segment features has better performance than that based on single-segment features by utilized the SVM, CART, and *k*-NN classifier. Compared with the single-segment features-based fault diagnosis models, multi-segment features-based fault diagnosis models have more features. However, in our opinion, the more important reason is that the multi-segment features reflect the potential information between different single segments.

## 5. Discussion

As mentioned in the introduction, in order to construct a stable and efficient fault diagnosis of PMDCMs, this paper presents a new multi-segment feature extraction method based on the time domain features. As we can see in Table 10, it can be found that the presented multi-segment feature extraction method performs better than the single-segment feature extraction method on the SVM, CART, and *k*-NN, respectively. A possible explanation for this might be that multi-segment features not only reflect the signals’ time domain feature but also reflect the correlation between different segments. The multi-segment feature extraction method may not be suitable for all kinds of signals; in other words, the multi-segment feature extraction method suitable for these signals, which have large fluctuations in different time periods. The proposed multi-segment feature extraction method provides a reliable fault diagnosis of PMDCMs.

In future research, frequency domain and time-frequency domain features should be considered. Feature dimensionality enhancement and feature dimensionality reduction methods should be taken into account, such as principal component analysis (PCA) and kernel principal component analysis (KPCA). The proposed multi-segment feature extraction method is not limited to the time series history of 2 s, which can be optimized according to the raw signal, and not limited to fault diagnosis, which can be utilized in pre-diagnosis. In the future work, optimization of the interval and number of single segments for the time series not being a stationary signal will be a significant study. Deep learning (DL) will be utilized for fault diagnosis of PMDCMs, such as convolutional neural network (CNN), recurrent neural network (RNN), and transfer learning. In addition, the combination of DL and attention mechanisms will also be carried out for further improving the fault diagnosis effect of PMDCMs.

## 6. Conclusions

In this paper, a novel multi-segment feature extraction method has been presented, which is utilized for fault diagnosis of PMDCMs. The main conclusions are as follows:(1)Multi-segment features-based fault diagnosis models have better performance than single-segment features-based fault diagnosis models.(2)A combination of the grid-search method and $k^{\prime }$-fold cross-validation is helpful to determine the parameters of the SVM.(3)Normally, the SVM-based model has better fault diagnosis performance than the CART-based and *k*-NN-based models for fault diagnosis of PMDCMs.

The effectiveness of the presented multi-segment feature extraction is validated through the fault diagnosis experiment of PMDCMs. This paper provides theoretical guidance for multi-segment feature extraction and fault diagnosis of PMDCMs.

## Figures and Tables

**Figure 1 sensors-21-07505-f001:**
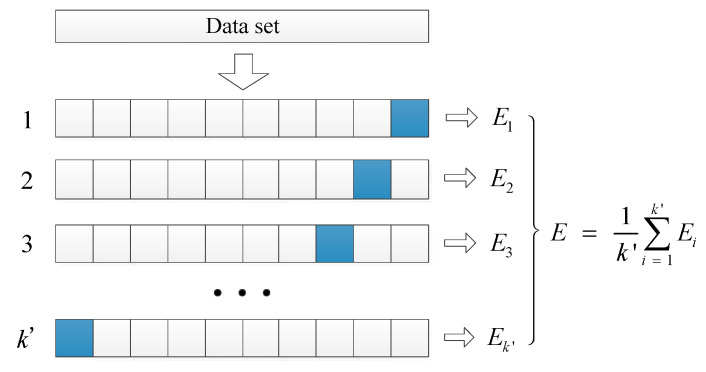
Procedure of $k^{\prime }$-fold cross-validation.

**Figure 2 sensors-21-07505-f002:**
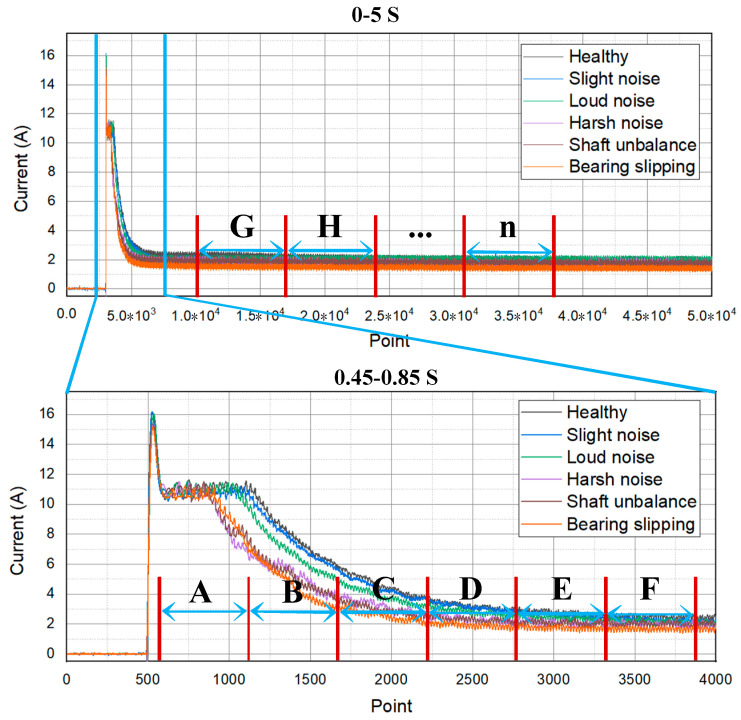
Multi-segment feature extraction.

**Figure 3 sensors-21-07505-f003:**
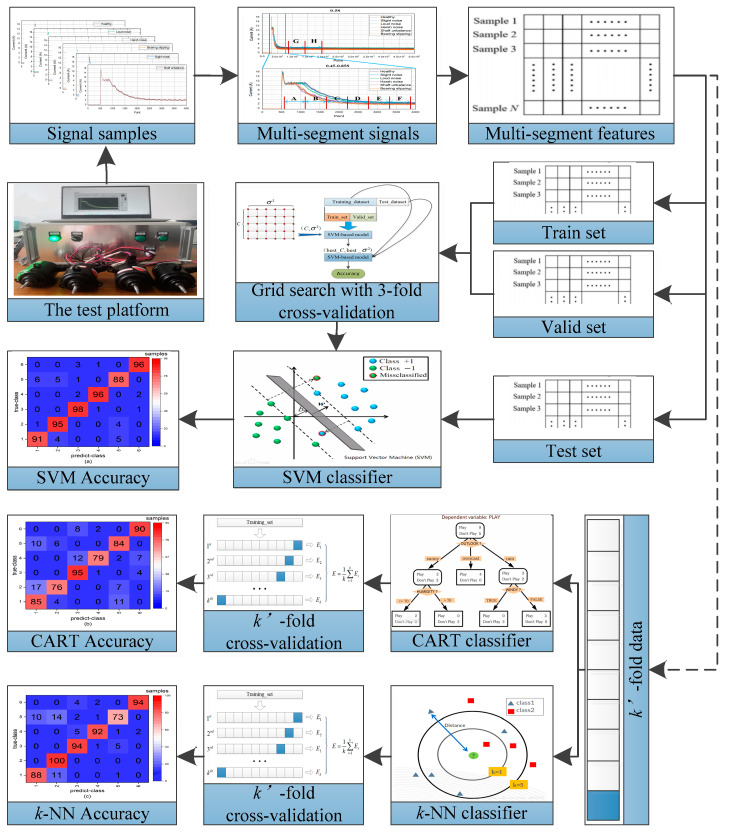
Fault diagnosis procedure of PMDCMs based on the SVM, CART, and *k*-NN by multi-segment features.

**Figure 4 sensors-21-07505-f004:**
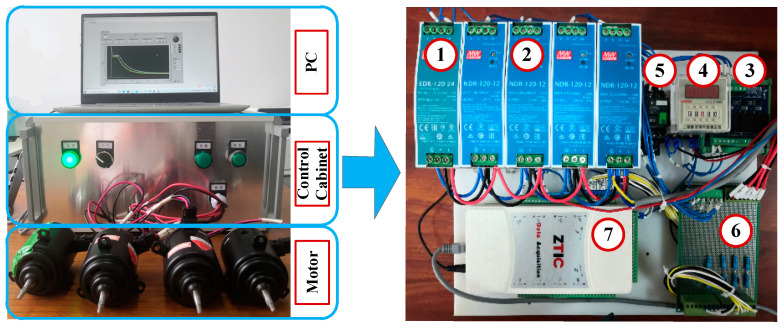
The fault diagnosis platform for PMDCMs.

**Figure 5 sensors-21-07505-f005:**
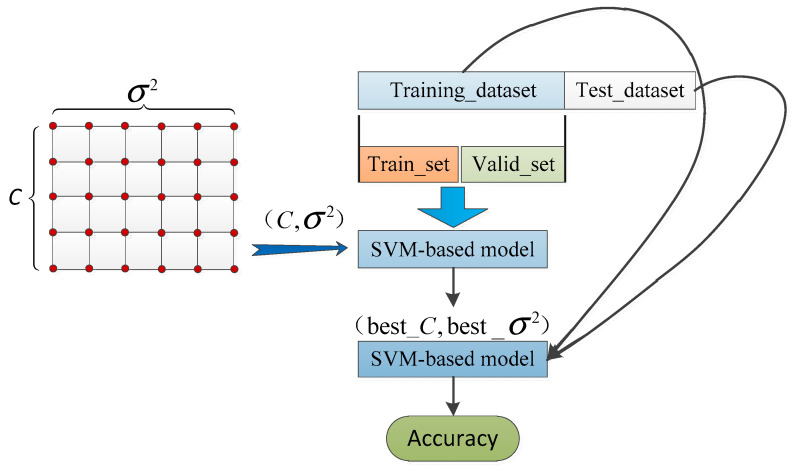
Parameter optimization of SVM for fault diagnosis of PMDCMs.

**Figure 6 sensors-21-07505-f006:**
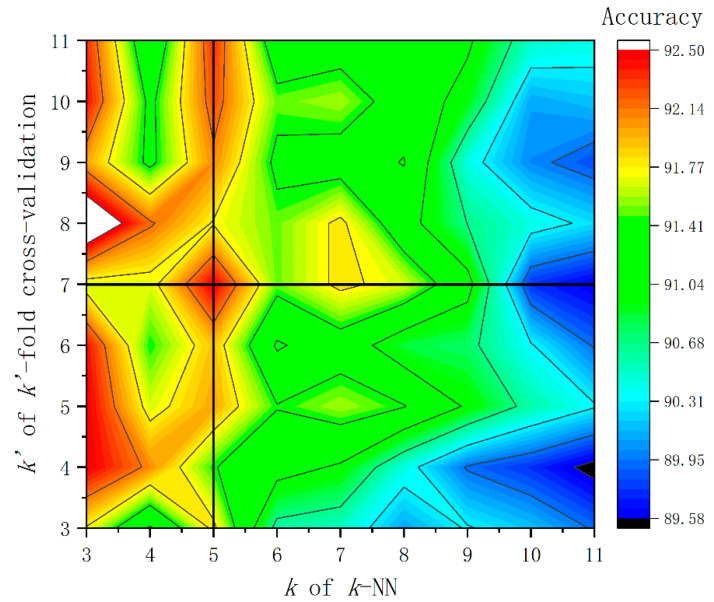
The results of *k*-NN-based fault diagnosis model utilizing the grid-search method.

**Figure 7 sensors-21-07505-f007:**
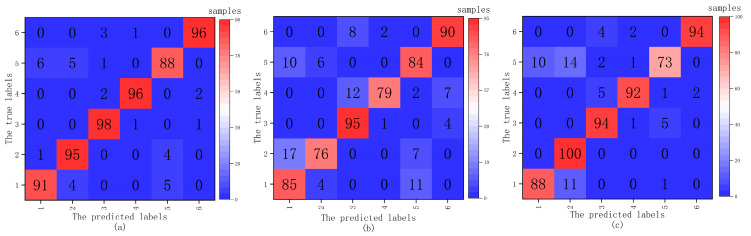
Confusion matrices of the three fault diagnosis models ((**a**) SVM-based model, (**b**) CART-based model, (**c**) *k*-NN-based model). The *x*-axis corresponds to the predicted labels, while the *y*-axis represents the true labels.

**Table 1 sensors-21-07505-t001:** The extracted time domain features.

No.	Abbreviation	Features	Expression
1	F1	Mean (μ)	$\mu =E\left(x_{i}\right)$
2	F2	Root Mean Square (RMS)	$x_{RMS}={\left\{E\left(x_{i}^{2}\right)\right\}}^{1/2}$
3	F3	Maximum (Max)	$x_{Max}=max\left(x_{i}\right)$
4	F4	Peak to Valley (PV)	$x_{PV}=\max\left(x_{i}\right)-min\left(x_{i}\right)$
5	F5	Standard deviation (Std)	$x_{Std}=\sigma ={\left\{E\left[{\left(x_{i}-\mu \right)}^{2}\right]\right\}}^{1/2}$
6	F6	Skewness (Ske)	$x_{Ske}=E\left\{{\left[\left(x_{i}-\mu \right)/\sigma \right]}^{3}\right\}$
7	F7	Kurtosis (Kur)	$x_{Kur}=E\left\{{\left[\left(x_{i}-\mu \right)/\sigma \right]}^{4}\right\}$
8	F8	Shape Factor (SF)	$x_{SF}=x_{RMS}/\mu$
9	F9	Crest Factor (CF)	$x_{CF}=x_{Max}/x_{RMS}$

**Table 2 sensors-21-07505-t002:** The strategies of multi-segment feature extraction.

	Duration 1	Segment No. 1	Duration 2	Segment No. 2	Feature No.
S0025	0.025 s	12	0.25 s	4	144
S005	0.05 s	6	1 s	2	72
S0018	0.1 s	8	0	0	72
S0028	0.2 s	8	0	0	72
S0038	0.3 s	8	0	0	72
S0048	0.4 s	8	0	0	72
S0058	0.5 s	8	0	0	72

**Table 3 sensors-21-07505-t003:** Type and function of the apparatuses in the control cabinet.

No.	Name	Type	Function
1	24 V power supply	Mean Well EDR-120-24	220 V AC→24 V DC
2	12 V power supply	Mean Well NDR-120-12	220 V AC→12 V DC
3	Front-end relay	BMZ 02R1-E	Switch of the whole circuit
4	Timer relay	ANLY AH3-2	Trigger delay: 0~10 s
5	4-channel relay	Schneider Electric RX4AB2BD	Switch of the motors
6	Sampling resistance	TO247-100W-0.2	High-frequency-based digital sampling
7	Data acquisition card	ZTIC EM9118B	Collection of current signals

Note: AC represents alternating current, DC represents direct current.

**Table 4 sensors-21-07505-t004:** Categorization for healthy and faulty PMDCMs.

Class	Actual Status	Number of PMDCMs
1	Healthy	3
2	Slight noise	1
3	Loud noise	2
4	Harsh noise	1
5	Shaft unbalance	1
6	Bearing slipping	1

**Note:** PMDCM represents permanent magnet DC motor.

**Table 5 sensors-21-07505-t005:** The performances of different multi-segment feature extraction strategies.

	S0025	S005	S0018	S0028	S0038	S0048	S0058
SVM	90.67	93.83	86.5	89.5	85	82.33	85.83
CART	89.41	89.16	86.25	84.75	79.59	77.75	77.42
k-NN	92.08	92.5	89.33	90.67	87.08	84.4	83

**Table 6 sensors-21-07505-t006:** The *k*-NN-based testing results for multi-segment features the S005 features).

	*k*	3	4	5	6	7	8	9	10	11
$k^{\prime }$										
3		91.75	91	91.83	90.58	90.5	90.08	90.33	90.17	89.92
4		92.5	92.08	91.42	91.17	91	90.42	89.92	89.75	89.5
5		92.5	91.67	92	91.42	91.58	91.42	91	90.58	90.33
6		92.42	91.42	91.92	91	91.25	90.83	90.75	90.33	89.92
7		91.66	91.67	92.5	91.5	91.83	91.67	91.16	89.75	89.58
8		92.92	92.17	91.75	91.5	91.83	91.17	90.67	90.42	90.25
9		92	91.33	92.08	91.33	91.25	91.42	90.41	90	89.83
10		92.42	91.33	92.25	91.5	91.58	91.25	90.92	90.08	90.17
11		92.34	91.08	92.34	91.17	91.25	91.34	91.09	90.5	90.42

Note: *k* represents the *k* of *k*-NN and $k^{\prime }$ represents the $k^{\prime }$ of $k^{\prime }$-fold cross-validation.

**Table 7 sensors-21-07505-t007:** The performances of models utilizing the F1 to F9 features set.

	F1	F2	F3	F4	F5	F6	F7	F8	F9
SVM	94	93	85	72.33	77.67	62.67	64.33	74.5	69.17
CART	88.42	89.16	79.33	58.41	69.58	49.58	55.58	69.83	60.84
k-NN	91.25	91.75	77.75	59.09	66.59	52.42	52.17	66.42	52.5

**Table 8 sensors-21-07505-t008:** The Pearson correlation coefficients between F2 and the other eight features.

	F1	F3	F4	F5	F6	F7	F8	F9
F2	0.9999	0.9342	0.2305	0.2474	0.2047	−0.2243	−0.395	−0.4567

**Table 9 sensors-21-07505-t009:** The performances of the model.

	S005	No_F1	No_F3	No_F1_F3
SVM	93.83	92.33	93.5	93.33
CART	89.16	90	90.45	90.41
k-NN	92.5	91.83	92.24	91

Note: No_F1 represents the features set, except F1; No_F3 represents the features set, except F3; and No_F1_F3 represents the features set, except F1 and F3.

**Table 10 sensors-21-07505-t010:** Testing results for single-segment features and multi-segment features.

		A	B	C	D	E	F	G	H	Ave	Multi	
	$C_{*}$	64	256	2	128	4	4	4	4		8	
SVM	$\sigma_{*}^{2}$	0.063	0.031	0.125	0.063	0.25	0.25	0.25	0.25		0.016	
	Acc	71.83	68.67	72.17	76.33	78.5	73.5	79.33	81.67	75.25	93.83	18.58
CART	Acc	68.83	68.83	68.08	72.75	69.59	69.91	73.67	72.92	70.57	89.16	18.59
*k*-NN	Acc	70.17	64.67	68.33	71.33	69.83	67.34	72.42	76.25	70.04	92.5	22.46
	Ave	70.28	67.39	69.53	73.47	72.64	70.25	75.14	76.95	71.95	91.83	19.88

Note: Acc represents accuracy, unit (%); Ave represents average; and Multi represents multi-segment features.

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
