# Peer review of "Fault Diagnosis of Permanent Magnet DC Motors Based on Multi-Segment Feature Extraction"

_sensors, 2021, doi:10.3390/s21227505_

Round 1
Reviewer 1 Report
The work of the paper is relatively complete as a whole, and the paper is also innovative. There are two main points that need to be improved. One is the basis of segmentation, how to determine the interval of segmentation? and the other is that the algorithm of fault diagnosis model is slightly complex and can be further simplified.
Reviewer 2 Report
I wonder if the formula in this article needs to be centered?
Reviewer 3 Report
The authors have presented a 'multi-segment feature extraction' for fault diagnosis in DC motors. They have used the extracted features as inputs for off-the-shelf machine learning models for fault classification. There are quite a few things of concern with the approach and the conclusions.
1) They have extracted multiple statistics from equally spaced 'segment' or windows in the current signal time series data. How is the window size chosen ? It has been heuristically set to 0.5 s. Is 0.5 s window size optimal ? As the time series is not stationary (clearly it has some transience at the beginning), what is the guarantee that 0.5 s would be sufficient for extracting statistics in each window?
2) Some of the statistics/features are very correlated, e.g. Standard deviation and root mean square. What is the need of 9 features? Have they tried a feature importance selector to select the required features? If there is a lower number of features that can provide comparable accuracy, training a more complicated model is not ideal.
3) I believe there is a fundamental lack of literature study/survey in the state-of-the-art in the field of Machine Learning to tackle such problems. For example, the authors have mentioned: "The raw signals are rarely adopted as the input of machine learning (ML) since they contain lots of redundant information". In deep learning models for time series classifications, e.g. Recurrent Neural Network architectures like LSTM networks, raw time series signals are indeed used as inputs, and the ML algorithm figures out the required features. So, it is not true that raw signals are 'rarely' used, in fact the authors can look at the last few years of work in the field of ML where researchers have shown tremendous success by using deep learning on raw data.
4) Fault diagnosis often falls under the purview of prognostic health monitoring whereby it is required to make very fast inference about the state of the system. The proposed framework relies on the entire time series history of 2s to extract features, and no consideration has been provided in the paper of how to reduce the data requirement for fault classification.
These are some of my major concerns about this work. In my opinion, this work is not publishable.
Reviewer 4 Report
Comments to the Author:
Comment 1: In this paper, a fault diagnosis model based on multi-segment feature extraction for PMDCM is designed. The diagnosis effect of three classifiers such as SVM after multi-segment feature extraction is only compared with traditional methods. The disadvantages of existing similar feature extraction methods are not explained, for example, the fault diagnosis based on MEEMD and SVM.
Comment 2: v-flod cross-validation was introduced in article 2.2. Is the k in Figure 1 in this section a misspelled v? In addition, there are many formatting errors in the article. Please correct them.
Comment 3: Please improving background and introduction by considering Data-driven fault diagnosis for traction systems in high-speed trains: A survey, challenges, and perspectives.
Comment 4: In the instructions for fault diagnosis steps, Step 6 divides all samples into equal length partitions on average. Would you mind indicating whether the determination criteria of V is related to the categorization for healthy and faulty PMDCMs?
Overall, the paper looks good, and a revision is necessary.
Round 2
Reviewer 3 Report
I do not think the authors have made the changes in the paper's methodology or formulation to address my comments. Almost all of the proposed modifications have been set aside as future investigation. I request the authors to make the necessary modifications.
Reviewer 4 Report
The reviewer appreciates the revisions from all authors. Now my suggestion is to Accept it.